# Women with Recurrent Pregnancy Loss More Often Have an Older Brother and a Previous Birth of a Boy: Is Male Microchimerism a Risk Factor?

**DOI:** 10.3390/jcm10122613

**Published:** 2021-06-14

**Authors:** Caroline Nørgaard-Pedersen, Ulrik Schiøler Kesmodel, Ole B. Christiansen

**Affiliations:** 1Centre for Recurrent Pregnancy Loss of Western Denmark, Department of Obstetrics and Gynaecology, Aalborg University Hospital, Reberbansgade 15, 9000 Aalborg, Denmark; u.kesmodel@rn.dk (U.S.K.); olbc@rn.dk (O.B.C.); 2Department of Clinical Medicine, Aalborg University, 9000 Aalborg, Denmark

**Keywords:** recurrent pregnancy loss, habitual abortion, reproductive immunology, microchimerism, Y chromosome

## Abstract

Known etiologic factors can only be found in about 50% of patients with recurrent pregnancy loss (RPL). We hypothesized that male microchimerism is a risk factor for RPL and aimed to explore whether information on family tree and reproductive history, obtained from 383 patients with unexplained RPL, was supportive of this hypothesis. The male:female sex ratio of older siblings was 1.49 (97:65) in all RPL patients and 1.79 (52:29) in secondary RPL (sRPL) patients, which differed significantly from the expected 1.04 ratio (*p* = 0.027 and *p* = 0.019, respectively). In contrast, the sex ratio of younger siblings was close to the expected ratio. Sex ratio of the firstborn child before sRPL was 1.51 (*p* = 0.026). When combined, 79.1% of sRPL patients had at least one older brother, a firstborn boy, or both. This differed significantly from what we expected based on the distribution of younger siblings and a general 1.04 sex ratio of newborns (*p* = 0.040). We speculate whether (s)RPL patients possibly acquired male microchimerism from older brother(s) and/or previous birth of boy(s) by transplacental cell trafficking. This could potentially have a detrimental impact on their immune system, causing a harmful response against the fetus or trophoblast, resulting in RPL.

## 1. Introduction

The most common pregnancy complication is miscarriage, which, in most cases, can be explained by serious structural malformations or chromosomal abnormalities which are incompatible with life [1], and 2–3% of fertile women suffer from recurrent pregnancy loss (RPL).

RPL of unknown etiology (uRPL) includes women with no uterine malformations, endocrine dysfunction or thrombotic disorders, and couples with no parental chromosome abnormalities, and this applies to about 50% of RPL patients. The identification of contributing risk factors and underlying causes of uRPL is essential in order to offer the most favorable support and, if possible, curative medical treatment.

Immunological disturbances have been suggested to play a pivotal role in the pathogenesis for uRPL, since autoantibodies [2,3], specific human leukocyte antigen (HLA) class II alleles [4], and immune imbalance of specific lymphocytes [5,6,7] are found more frequently in uRPL patients than in healthy controls. 

A potential immunological risk factor for RPL is microchimerism. Microchimerism is defined as small amounts of foreign cells or DNA detectable in a genetically distinct individual. 

A frequent source of microchimerism is the transplacental trafficking of cells through the placenta. As early as the first trimester, male-specific minor histocompatibility (HY)-specific T-cell responses were detected. Microchimeric cells have the potential to persist for many decades post-partum [8,9,10,11]. These cell grafts can potentially affect the womans’ immune tolerance to a pregnancy; however, we still question if these foreign cells act in a beneficial or detrimental manner, or both [10,12].

Microchimerism is not only acquired in the childbearing female, but also in her fetus, by fetomaternal cell trafficking across the placental in both directions. Male microchimerism has been found in nulliparous women; thus, in women with no history of miscarriages, suggested sources have been an unrecognized pregnancy loss, sexual intercourse, blood transfusion, a twin or a known or unknown vanished twin, or the microchimeric cells could derive from an older brother via the mother of the proband (MP) [13].

Microchimerism from the MP or the proband’s child(ren) has been investigated more thoroughly than microchimerism from other sources, such as older siblings. Microchimerism acquired from the proband’s child(ren) has been proposed as a risk factor for autoimmune diseases, as it is more prevalent in females [14,15,16]. In transplant recipients, the risk for graft versus host disease (GVHD) is higher when the donor is a parous rather than a nulliparous female [9,17]. Pathologic responses to semiallogeneic cells in uRPL patients have been suggested to resemble the immunological mechanisms and responses involved in autoimmune diseases and GVHD and, therefore, microchimerism has also been suggested as a risk factor for uRPL [12,18].

The carriage of HLA class II alleles able to present HY-antigens to maternal T-helper lymphocytes in women with secondary RPL (sRPL) after a first-born boy have been associated with a decreased chance of a live birth compared to similar sRPL patients without these alleles [18]. This supports the theory that the poorer prognosis in patients with RPL after the birth of a boy is due to an unfavorable immune response to HY-positive cells. A persistent fetal-specific immune response seems to be characterized by an effector memory T-cell phenotype that retained the ability to proliferate, secrete IFN-gamma, and lyse male target cells many years after pregnancy when stimulated in vitro [11]. Based on these studies, cell grafts are associated with detrimental actions in at least some women.

In this purely clinical study, we aimed to explore if a history of an older brother and/or a first-born boy was more frequent in RPL patients compared to the expected distribution, and if the association was more pronounced in primary (pRPL) or sRPL. If this is confirmed, it supports the hypothesis of a role for male microchimerism in the pathogenesis of RPL.

## 2. Materials and Methods

All patients who, from January 2016 to March 2021, were admitted to The Centre for Recurrent Pregnancy Loss of Western Denmark located at Aalborg University Hospital were included in the study if the study criteria were met. In all patients, hysterosalpingography, hysteroscopy or hydrosonography screening for uterine abnormalities were performed and blood samples were collected to screen for RPL risk factors, as is recommended by the European Society of Human Reproduction and Embryology (ESHRE) Guidelines on RPL [19]. Furthermore, in most couples, karyotyping was also done on peripheral blood.

Data for this study were collected retrospectively from medical records from all patients admitted to The Centre for Recurrent Pregnancy Loss. At the first consultation, in addition to a detailed recording of their medical and reproductive history, the women were asked to give information about their full and half siblings: their sex (biological attribute), age compared with the proband, and whether they were full siblings or half siblings. In the latter case, it was asked whether they were siblings on the maternal or paternal side, since only siblings sharing the same mother were considered relevant in the present study. Data on pregnancy losses in the MP before giving birth to the proband were not obtained since they would rarely, if ever, contribute information regarding the gender of these predominantly early pregnancy losses. The data were entered into the centre’s research database, from which data for this study were extracted.

Women were included if they had ≥3 consecutive pregnancy losses including both biochemical and clinical miscarriages. Verified complete molar and ectopic pregnancy losses were not counted in the number of losses. Exclusion criteria were uterine malformations (*n* = 8), adopted as child (*n* = 3), no family history obtained at first consultation (*n* = 16), having a twin (*n* = 3), patient or husband carrying a known significant chromosomal translocation (*n* = 4) and ≤2 consecutive pregnancy losses (*n* = 16). A total of 383 were included for analysis.

## 3. Statistical Analysis

Patients were divided into four groups according to the sex of older siblings with whom they shared the same mother: (1) patients who had only one or more older brothers, (2) patients who had only one or more older sisters, (3) patients who had both older brother(s) and sister(s) and (4) patients with no older siblings.

Data were analyzed in Stata/MP (StataCorp LCC. 2017. Stata Statistical Software: Release 15. College Station, TX, USA). Level of statistical significance was defined as *p* < 0.05. Differences in continuous parametric variables were compared using unpaired *t*-test, while continuous non-parametric variables were compared using two-sample Mann–Whitney U test. When comparing three groups or more on a non-parametric variable, Kruskal–Wallis H Test was used, while a one-way ANOVA was used for parametric variables. Categorical variables were compared using chi-squared test, while Fischer’s exact test was used when small numbers were expected in at least one group.

We tested for deviation of sex distribution from an expected 51%:49%, which equals a 1.04 male:female sex ratio in older siblings or firstborn child using binomial test, since this was the sex ratio of Danish newborns in 2017. In the binomial test, we only included patients with older siblings of one sex and excluded patients with no older siblings or older siblings of both sexes. We expected that the probability of having only older brother(s) equals the probability of having only older sister(s). This test was performed on all RPL patients and then stratified by RPL subgroups. When we reported the number of patients with at least one older brother or sister, the sexes of all the patient’s older siblings were accounted for. The same tests were performed in younger siblings and firstborn children. Regarding the sex ratio of firstborn children, only women with either only boy(s) or only girl(s) before RPL were included, and those with births of children of both sexes were not counted in this sex ratio. However, the sex of all children born >22 weeks of gestation before the RPL of all sRPL patients were included when analyzing the number of patients with at least one boy or girl. In total, 39 (21.4%) sRPL patients had ≥2 children before the sRPL diagnosis, including six patients who had three children and one patient who had four births before sRPL.

Next, we listed the number of patients with a minimum of one older brother, one birth of a boy before RPL, or both, observed in the present study, in contrast to the number with only older sister(s), no siblings and only birth of girls before RPL. The former group represents patients with a potentially increased chance of microchimerism of HY-positive cells due to the transfer of cells from older brothers via their mothers’ placentas or transfer of cells from their own previous pregnancies with boys. The latter group represents patients with a low chance of HY cell microchimerism, since neither their mothers nor themselves were exposed to ongoing pregnancies with fetuses and placentas expressing HY antigens (Figure 1).

We had no reference group of “normal” women to serve as a comparison group with regard to the frequency of at least one older brother and/or birth of a boy before RPL. However, we had details on our RPL patients’ younger siblings and the sex ratio of Danish newborns in 2017. We assumed that the frequency and the distribution of younger siblings according to sex would be a proxy for the frequencies and distribution of older siblings according to sex in a non-sRPL population. Using the data on younger siblings, 127 (69.8%) non-RPL women were expected to have had ≥1 older brother and/or ≥1 delivery of a boy.

This calculation is as follows: we observed that 56 out of 182 parous (sRPL) patients had at least one younger brother, and 126 (87 + 39) parous (sRPL) patients had no younger siblings or only younger sister(s) (Table 1 and Figure 2). We assume that the distribution of younger siblings represents the distribution of older siblings in “normal” women. Among the 126 parous women with no older brother(s), we expected that 51% had given birth to a boy in each of their pregnancies. Based on the frequency of ≥2 births before sRPL observed in our sRPL sample, 27 (21.4%) women with no older brother(s) were expected to have given birth to ≥2 children. The calculations of the expected/hypothetical probability of having delivered ≥1 boy before RPL are shown in Figure 2. We expected that 64 (50 with 1 boy + 14 with ≥2 boys) parous women with no older brother(s) would only have given birth to boys and 7 women would have both boy(s) and girl(s). This calculation showed that 56 + 64 + 7 = 127 (69.8%) sRPL patients were expected to have had older brother(s) and/or delivered ≥1 boy before RPL.

The same analysis was performed on all RPL patients; thus, we added all pRPL patients. Among the 201 pRPL patients, 69 patients had ≥1 younger brother while 132 pRPL patients had only younger sister(s) or no younger siblings. Therefore, we expected 189 RPL patients to have had an older brother and/or delivered a boy before RPL.

## 4. Results

The study sample comprised 383 RPL patients, including 182 (47.5%) sRPL patients who had a previous childbirth >22 weeks. In total, 202 (52.6%) RPLs patients had ≥1 older full siblings or maternal half siblings and 213 (55.6%) had ≥1 younger full siblings or maternal half siblings (Table 1). 

Comparing baseline characteristics between the four subgroups of RPL patients: patients with only older brother(s), patients with only older sister(s), patients with older siblings of both sexes and patients with no older siblings, showed no significant differences (Table 2). Comparing patients with only older brother(s) with patients with only older sister(s) showed no differences either.

### 4.1. Frequency of an Older Brother and Sister

Among the 383 RPL patients. 25.3% had only older brother(s) and 17.0% had only older sister(s), which corresponds to a male:female sex ratio of 1.49 (Table 1). The male:female sex ratio differed significantly from the expected 1.04 sex ratio among siblings of all RPL patients (*p* = 0.027) and among siblings of sRPL patients (*p* = 0.019). More pRPL patients had only older brother(s), but the sex ratio of 1.25 was not significant. The sex ratio in younger siblings of all RPL patients was 0.86, which did not differ from the expected 1.04 ratio, and the same was true for the sex ratio of younger siblings in RPL subgroups.

Among all RPL patients with a minimum of one older brother, 113 (81.3%) had one older brother, 16 (11.5%) had two older brothers, 8 (5.8%) had three older brothers and 2 (1.4%) had five older brothers. Among all patients with a minimum of one older sister, 86 (79.6%) had one older sister, 17 (15.7%) had two older sisters and 5 (4.8%) had three older sisters.

### 4.2. Age Difference

The age difference between the proband and her youngest older brother and youngest older sister is illustrated in Table 3. The age difference between the youngest older brother and youngest older sister did not differ in all RPL patients or in RPL subgroups. There was no significant difference in age difference from proband to her youngest older brother when comparing pRPL with sRPL patients. Furthermore, the age difference regarding the youngest older sister in pRPL patients was comparable to the age difference in sRPL patients.

### 4.3. Sex Ratio of Previous Birth

In the probands with sRPL, the frequency of previous birth of only boys and only girls was skewed, showing a significantly higher frequency of previous birth of a boy compared to the expected 1.04 ratio. The sex ratio was 1.51 (95:63, *p* = 0.026). The sex ratio of firstborns was even higher in sRPL patients with older sisters only (2.50); this ratio was, however, not significant (Table 4). Among sRPL patients with only older sisters, 72.4% had previously given birth at least one boy, 31.0% had given birth at least one girl, and 3.4% had given birth to a girl and a boy before sRPL. The sex ratio of older siblings to pRPL patients was 1.25, while, in sRPL after girls only or boys only, the sex ratio of older siblings was 1.67 and 1.54, respectively. These sex ratios did not differ significantly.

### 4.4. Combining Sex of Older Siblings and First-Born Child(ren)

Of all sRPL patients, 79.1% had at least one older brother or had given birth to a boy before the RPL diagnosis, or both, while 20.9% had only older sister(s), no older siblings and/or a firstborn girl and, thus, no known exposure to HY-antigens (Table 5).

The question is whether this 79.1% versus 20.9% distribution is different from that expected from the background population. As described in the statistical section and Figure 2, we expected that 127 (69.8%) women would have an older brother and/or previous birth of a boy. Compared with the observed number of 144 (79.1%) sRPL with the same family/reproductive history, the difference was statistically significant (*p* = 0.041, Table 5).

When we performed the same analysis in all RPL patients, the difference was no longer significant (Table 6).

## 5. Discussion

The present study has, for the first time, provided data suggesting that events in the MP’s pregnancies prior to the birth of the proband herself are risk factors for the development of sRPL. Previous studies have suggested that birth of a boy before sRPL is significantly more frequent than expected, and that it negatively impacts the subsequent pregnancy prognosis [20,21]. Subsequent follow-up studies suggested that this reduction in reproductive fitness was due to immunity against HY-antigens, expressed on fetal or trophoblast cells [18,22]. The time interval between the birth of the youngest older brother and the proband seemed not to impact the risk of sRPL (Table 3). Thus, the influence of male microchimerism transferred via the mother to younger siblings does not appear to be short-lived.

In the present study, significantly more (s)RPL patients had older brother(s) compared with the expected number, and there was also a significant excess of firstborn boys born before the sRPL diagnosis, which confirms previous findings [20,21]. In an analysis combining these two observations, 79.1% of sRPL patients either had an older brother, had given birth to a boy before the RPL diagnosis, or both. This was significantly higher than the expected frequency (69.8%) of women with a similar family history and history of prior deliveries in the background population based on the observed frequency and sex distribution of the patients’ younger siblings in our sample and an expected 1.04 sex ratio in the population. Based on these findings, we propose that sRPL patients with older brother(s) and/or birth of boy(s) have a high probability of acquiring male microchimerism, supporting the hypothesis that immunity against HY-antigens plays a role in the pathogenesis of sRPL.

The sex ratio of older siblings among patients with pRPL was lower than the sex ratio found among patients with sRPL with a previous birth of a girl. Although the difference was non-significant, the finding may initially seem unexpected, since the pRPL patients have (probably) not been previously exposed to male microchimeric cells in their own pregnancies and, thus, an increased prevalence of older brothers could be expected if anti-HY immunity plays a role in the pathogenesis of RPL. However, meta-analyses of randomized controlled trials of immunotherapy with intravenous immunoglobulins (IVIG) in RPL patients seem to indicate that this kind of immunotherapy only works in sRPL patients and not in pRPL patients [23,24]. This could support the hypothesis that the pathogenesis of pRPL is often non-immunological (anatomical, genetic, endocrine and hemostatic alterations), which is not benefited by immunotherapy. This is in accordance with the findings of this study, which suggest that a substantial proportion of sRPL cases is associated with anti-HY immunity: an alloimmune disorder that may be modified by immunotherapy with IVIG-like alloimmune thrombocytopenia [25].

The transplacental trafficking of cells from the fetus to the mother and the reverse is now well-documented as occurring during a normal human pregnancy. Fetal cells entering the maternal circulation have been documented to be able to survive in the maternal organism for decades and establish the so-called microchimerism [8,16,26]. According to previous studies, the level and frequency of male microchimerism were significantly higher in women who had experienced an induced abortion [13,27]. These fetal cells may interact with the maternal cellular immune system and cause lifelong immunological changes: both the induction of immunological tolerance and harmful immunity against the semiallogeneic cells are possible.

Microchimerism may be established over generations. Fetal cells entering the MP circulation during the first pregnancy may persist and represent a preexisting inhabitant that, in a subsequent pregnancy, may cross the placenta and establish microchimerism in the second child (Figure 1). The second-born child will consequently acquire long-lasting or permanent microchimerism originating from both the MP and an older sibling. A pregnancy loss in the MP before giving birth to the proband is also a possible source of microchimerism. This information was not possible to collect, since the patients rarely knew if their mother had prior pregnancy losses and, in addition, the sex of these predominantly early pregnancy losses is rarely known. Therefore, this information would not add any information to this preliminary study beyond what we already know. It is a minor but potential source of microchimerism that may explain why some women with no older brother or previous birth of a boy harbor male microchimeric cells.

The acquired microchimerism from the MP during intrauterine life may stimulate the fetal T-cells and induce Treg differentiation with suppressive a function that persists into adulthood [28]. This acquired tolerance to maternal microchimerism was theorized to occur against any antigens encountered in the maternal circulation while in utero; thus, male microchimeric cells in the MP, which may have been acquired from a previous pregnancy with a male fetus, cross the placenta into the fetus (the probands in this study) during a subsequent pregnancy, and could cause the differentiation of long-lived Tregs specific to HY-antigens acquired from an older brother via the mother. Gammill et al. [12] hypothesized that the microchimerism acquired from the MP possibly plays an active role in the proband’s capacity to adapt to future pregnancies; thus, it is of particular relevance in patients with RPL.

Dierselhuis et al. [29] discovered male microchimerism in six of nine umbilical cord blood (UCB) samples from newborn female infants with older brothers while, in contrast, male microchimerism was found in none of the UCB samples from females with no older brothers. Additionally, hepatocytes containing XY chromosomes in liver biopsies from female fetuses and children have been reported [26]. Furthermore, several studies have observed a lower risk of acute and chronic GVHD when younger siblings were used as donors in allogeneic stem cell transplantation (SCT) compared to older siblings. This suggests that the younger siblings were possibly exposed to cells from the older siblings, inducing B- and T-cell sensitization and the differentiation of Treg cells, which may prevent lymphocyte activation after SCT [30,31].

Thus, these findings strongly support our hypothesis that microchimerism acquired from older siblings in the MP can be passed on to younger siblings and is a normal phenomenon, occurring in the pregnancies of multiparous women. As a consequence, these allogeneic cells from MP and older siblings can be harbored in the younger sibling until adulthood, or even remain lifelong, and affect his/her immune system.

Nonetheless, while such transplantation studies suggest a beneficial effect of developing tolerance to older siblings, other studies have suggested harmful impacts of microchimerism from older siblings and previous births in some conditions. One example is systemic sclerosis (SSc), as a study observed an increased risk of developing SSc with an increasing number of older siblings and a history of pregnancy loss with or, especially, without a live birth [32]. The risk of developing systemic lupus erythematosus (SLE) also increases with birth order [33]. In agreement with these autoimmune diseases, the present study also suggests that, in women with sRPL, male microchimerism may result in harmful rather than beneficial immunological responses to HY antigens.

The results of this study have limitations and must be considered preliminary. We aimed to test if RPL patients had an older brother more frequently than expected. The estimation of how frequently we could expect RPL patients to have at least one older brother was based on the observed frequency of having a least one younger brother (Table 1), since the distribution of sex of siblings before and after the birth of the proband should, theoretically, be similar if the male microchimerism did not exist and did not impact the risk of sRPL. The almost equal number of older and younger siblings allowed us to use the frequency of younger brothers and sisters in this context. In a follow-up study, it would strengthen the conclusions if an external control group, for the distribution of older and younger siblings according to sex in a group of women with normal fertility, could be collected.

Our study could not document whether sRPL patients with an older brother actually received male microchimeric cells from that older brother or not. Therefore, we can only speculate that the high prevalence of older brothers among sRPL patients could indicate that microchimerism from older brothers, acquired via maternal circulation, could be a risk factor for sRPL. To confirm our hypothesis, we suggest that new studies should analyze if microchimeric cells from an older brother can be detected in sRPL patients more often than in women without RPL.

## Figures and Tables

**Figure 1 jcm-10-02613-f001:**
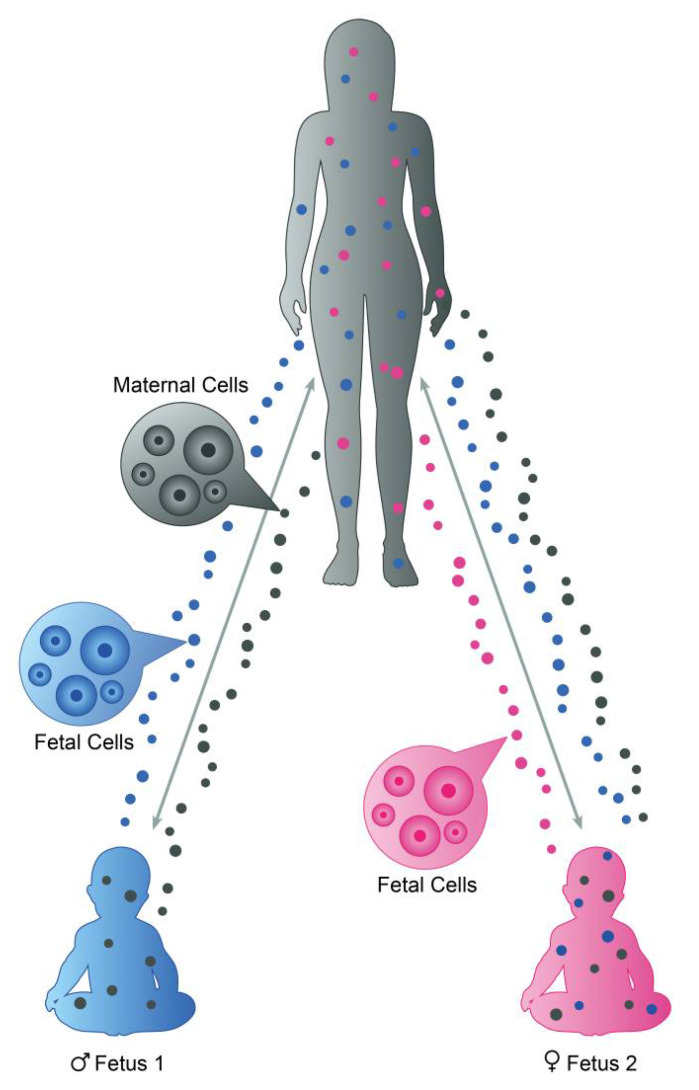
The mother of proband acquires fetal cells during her first pregnancy (fetus 1) which cells are transferred to fetus 2 in a subsequent pregnancy. The proband with RPL is illustrated as fetus 2 in this figure; she harbors microchimeric cells from both her mother and her older brother.

**Figure 2 jcm-10-02613-f002:**
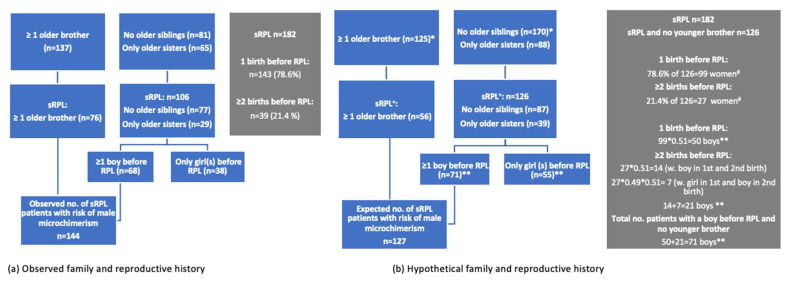
(**a**) The observed distribution of older siblings and children born before RPL according to sex to the left and (**b**) the hypothetical distribution of the same parameters to the right, which was based on distribution of the younger siblings and a 1.04 male:female sex ratio in newborns. The blue boxes illustrate how we calculate the observed and hypothetical number of women with an older brother and/or prior birth of a boy. The grey boxes illustrate (**a**) the observed distribution of 1 and ≥2 children born before RPL and (**b**) the calculation of the hypothetical number of women with birth of ≥1 boy used in the blue boxes, an estimate based on the observed distribution of 1 or ≥2 children. * Calculation based on frequency of having at least one younger brother (left upper box), or only sister(s) and no younger siblings among sRPL patients (right upper box) (see Table 1). ^+^ Among the 182 patients, 56 had at least one younger brother, while 126 had only sister(s) or no younger siblings. ^#^ Calculation based on the observed distribution of 1 (78.6%) or ≥2 children (21.4%) before RPL. ** Calculation based on the known 1.04 sex ratio of newborn children in the Danish background population.

**Table 1 jcm-10-02613-t001:** Frequency of older siblings stratified according to sex in all RPL patients and in primary RPL (pRPL) and secondary RPL (sRPL) patients.

	All RPL Patients (*n* = 383)	pRPL (*n* = 201)	sRPL (*n* = 182)
No older siblings, *n* (%)	181 (47.3)	104 (51.7)	77 (42.3)
Older siblings of both sexes, *n* (%)	40 (10.4)	16 (8.0)	24 (13.2)
Only older brother(s), *n* (%)	97 (25.3) ^a^	45 (22.4)	52 (28.6) ^b^
Only older sister(s), *n* (%)	65 (17.0) ^a^	36 (17.9)	29 (15.9) ^b^
At least one older brother, *n* (%)	137 (35.7)	61 (30.4)	76 (41.8)
At least one older sister, *n* (%)	105 (27.4)	52 (25.9)	53 (29.1)
Only younger brother(s), *n* (%)	76 (19.8)	42 (21.0)	34 (18.7)
Only younger sister(s), *n* (%)	88 (23.0)	48 (24.0)	39 (21.4)
Younger siblings of both sexes, *n* (%)	49 (12.8)	27 (13.5)	22 (12.1)
No younger siblings, *n* (%)	170 (44.4)	83 (41.3)	87 (47.8)
At least one younger brother, *n* (%)	125 (32.6)	69 (34.3)	56 (30.8)
At least one younger sister, *n* (%)	137 (35.8)	76 (37.8)	61 (33.5)

^a^: Binomial test compared to expected 1.04 sex ratio. *p* = 0.027; ^b^: Binomial test compared to expected 1.04 sex ratio. *p* = 0.019.

**Table 2 jcm-10-02613-t002:** Demographic data on recurrent pregnancy loss (RPL) patients stratified according to the sex of older siblings. All demographic data were obtained at time of referral.

Demographic	Only Older Brother(s) (*n* = 97)	Only Older Sister(s) (*n* = 65)	Older Siblings of Both Sexes (*n* = 40)	No Older Siblings (*n* = 181)
Age, years Mean (SD)	33.3 (5.6)	32.6 (5.2)	32.4 (5.8)	32.8 (5.1)
No. of consecutive losses, Median (range)	4 (3;12)	4 (3;13)	4 (3;10)	4 (3;12)
BMI, kg/m^2^ Mean (SD)	26.1 (5.4)	26.5 (6.5)	26.2 (5.8)	26.2 (5.7)
Smoking, %	13.4	18.5	5.0	9.4

**Table 3 jcm-10-02613-t003:** Age difference from the proband to her youngest older sibling in all RPL patients and in pRPL and sRPL patients.

	All RPL	pRPL	sRPL
Age difference to the youngest older brother (year) Median (P10:P90)	3.5 (1.5:7.0)	3.0 (1.5:7.0)	4.5 (2.0:7.0)
Age difference to the youngest older sister (year) Median (P10:P90)	3.5 (2.0:7.0)	3.5 (2.0:5.0)	3.5 (1.5:7.0)

P10:P90: 10th and 90th percentiles.

**Table 4 jcm-10-02613-t004:** Sex ratio of the firstborn child in sRPL patients stratified according to sex of RPL patient’s older siblings.

	Only Older Brother(s)	Only Older Sister(s)	Both Older Brother(s) and Sister(s)	No Older Siblings	At Least One Older Brother	All sRPL
Sex ratio of births >22 weeks ^a^	1.33	2.50	2.00	1.23	1.48	1.51 ^b^
Minimum one boy, *n* (%) ^c^	34 (65.4)	21 (72.4)	17 (70.8)	47 (61.0)	51 (67.1)	119 (65.4)
Minimum one girl, *n* (%) ^c^	28 (53.9)	9 (31.0)	10 (41.7)	40 (52.0)	38 (50.0)	87 (47.8)

^a^: Only RPL patients with birth of boy(s) or girl(s) only were included. ^b^: Binomial test compared to expected 1.04 sex ratio, *p* = 0.026. ^c^: % of sRPL patients.

**Table 5 jcm-10-02613-t005:** The observed and expected distribution of siblings and previous births in all sRPL patients according to sex. All previous births before sRPL were included (*n* = 182).

	Observed		Expected *	
	Frequency	Percentage (%)	Frequency	Percentage (%)
Older brother and/or a prior boy	144	79.1	127	69.8
No older brother and no prior boy	38	20.9	55	30.2

* Calculations based on assumptions in statistics section and Figure 2. χ^2^-test, *p* = 0.041.

**Table 6 jcm-10-02613-t006:** The observed and expected distribution of siblings and previous births in all RPL patients according to sex. All previous births before sRPL were included (*n* = 383).

	Observed		Expected *	
	Frequency	Percentage (%)	Frequency	Percentage (%)
Older brother and/or a prior boy	205	53.5	189	49.0
No older brother and no prior boy	178	46.5	194	51.0

* Calculations based on assumptions in statistics section and Figure 2. χ^2^-test, *p* = 0.247.

## Data Availability

The data presented in the present study is available on request from the corresponding author (C.N.-P.).

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
