# Peer review of "Women with Recurrent Pregnancy Loss More Often Have an Older Brother and a Previous Birth of a Boy: Is Male Microchimerism a Risk Factor?"

_jcm, 2021, doi:10.3390/jcm10122613_

Round 1
Reviewer 1 Report
The authors investigate the possibility of male microchimerism presenting a risk factor in recurrent pregnancy loss with said male michrochimerism resulting either from an older male sibling or, in the case of sRPL, a previous birth of a boy.
The introduction is well-written, providing sufficient background while including several relevant references for the reader.
In the materials and methods, the authors describe the very clear inclusion and exclusion criteria, while the extensive statistical analysis describes as clearly as possible the reasoning behind the way the analysis was performed. Since I am not familiar with the Stata/MP statistical software, I can not comment on that aspect.
In page 3, line 141-148 the authors explain how in the lack of a group with “normal” women they calculated the expected frequencies in non-RPL women. This can weaken in my opinion the study, however in page 9, lines 319-328 the authors themselves acknowledge this fact and further defend their strategy. As a “preliminary” study, this is for me accepted.
The results are in their most part clearly presented and interpreted. However, in my opinion not enough attention is given in the pRPL group. In the abstract, page 1, lines 14-16, the authors mention that “Male:female sex ratio of older siblings was 1.49 (97:65) in all RPL patients and 1.79 (52:29) in secondary RPL (sRPL) patients, which differed significantly from the expected 1.04 ratio (p=0.027 and p=0.019, respectively)” and in page 6, table 2 the authors offer the corresponding values for the pRPL group which, if I am not mistaken, result in a ratio of 1.25 (45:36). Is this difference also significant? My concern is the following, in the sRPL group male microchimerism can be a result of an older male sibling, a previous birth of a boy or a combination of both. However, in the pRPL group, male microchimerism can only be a result of an older male sibling. Therefore, one would expect that if male microchimerism resulting from an older male sibling were an RPL risk factor, the frequency of male siblings in the pRPL group should hypothetically be much higher than in the sRPL group (which is not) and equal to those in the sRPL sub-group of women with previous births of only girls. I would highly encourage the authors to comment on this fact and offer an explanation.
Finally, the conclusions presented in the discussion are, in my opinion, supported from the results. Furthermore, the authors highlight the value of their results, without failing to note the shortcomings and limitations of the study, while at the same time offer suggestions for future project in this subject.
Minor concerns:
In page 2, lines 51-55 the sentence is not very clear at the end. I would suggest to change it to “sexual intercourse, blood transfusion, a known or unknown vanished twin, a twin, or from an older brother via the mother of the proband (MP)”, or improve it in any way they see fit.
In page 3, line 100, the “and” is confusing, and the authors should consider changing the sentence to “Exclusion criteria were uterine malformations (n=8), adopted as child (n=3), no family history obtained at first consultation(n=16), having a twin (n=3), patient or husband carrying a known significant chromosomal translocation (n=4) ), and ≤2 consecutive pregnancy losses (n=16)”.
In page 7, Table 4.a. the authors should consider changing the legend to “The observed and expected distribution of siblings and previous births in all sRPL patients according to sex. All previous births before sRPL were included (n=182)” thus being more clear and consistent with the legend of Table 4.b.
Lastly, I encourage the authors to comment in the material and methods the possibility of a miscarriage or abortion of the proband’s mother, which occurred before the proband’s birth and that the proband is unaware of.
Overall, I find the manuscript extremely well written and quite interesting to read.
Author Response
Dear Reviewer 1.
Thank you for your review report. I am pleased to see you find the study interesting. I find your suggestions very relevant and I hope that the changes we have made fulfill what you suggested to be changed in the manuscript.
The manuscript now include a more detailed description and discussion of the findings in the pRPL patients. Previous studies has suggested that immunological disturbances may play a more pivotal role in sRPL than pRPL, and the lower sex ratio of older siblings among pRPL support this theory.
All of your minor concerns has been changed according to your suggestions.
Best regards,
Caroline Nørgaard-Pedersen
Reviewer 2 Report
The present study gives a new insight regarding RPL pathophysiology in terms of male microchimerism with results being supported with a satisfying sample size.
Minor spell check is required (e.g. the word "trafficking").
(Figure 3) in "Discussion" session should be removed.
Lines in Table 3 should align.
Author Response
Dear Reviewer 2
Thank you for your review report. I am pleased to see you find the study interesting. I find your suggestions very relevant and I hope that the changes we have made fulfill what you have suggested to be changed in the manuscript.
We have checked the manuscript for spelling errors and hopefully corrected all of them.
Figure 3 has been removed and age differences to the youngest older sibling is now illustrated in a Table 3. We hope that you will find this appropriate.
Lines in table 3 should now be align.
Best regards,
Caroline Nørgaard-Pedersen